# Computational Design and Development of Benzodioxane-Benzamides as Potent Inhibitors of FtsZ by Exploring the Hydrophobic Subpocket

**DOI:** 10.3390/antibiotics10040442

**Published:** 2021-04-15

**Authors:** Valentina Straniero, Victor Sebastián-Pérez, Lorenzo Suigo, William Margolin, Andrea Casiraghi, Martina Hrast, Carlo Zanotto, Irena Zdovc, Antonia Radaelli, Ermanno Valoti

**Affiliations:** 1Dipartimento di Scienze Farmaceutiche, Università degli Studi di Milano, Via Luigi Mangiagalli, 25, 20133 Milano, Italy; lorenzo.suigo@unimi.it (L.S.); a.casiraghi.89@gmail.com (A.C.); ermanno.valoti@unimi.it (E.V.); 2Centro de Investigaciones Biológicas (CSIC), Ramiro de Maeztu 9, 28040 Madrid, Spain; victorsebastianperez@gmail.com; 3Exscientia, The Schrödinger Building, Oxford Science Park, Oxford OX4 4GE, UK; 4Department of Microbiology and Molecular Genetics, McGovern Medical School, University of Texas, Houston, TX 77030, USA; William.Margolin@uth.tmc.edu; 5Faculty of Pharmacy, University of Ljubljana, Aškerčeva cesta, 7, 1000 Ljubljana, Slovenia; martina.hrast@ffa.uni-lj.si; 6Dipartimento di Biotecnologie Mediche e Medicina Traslazionale, Università degli Studi di Milano, Via Vanvitelli, 32, 20129 Milano, Italy; carlo.zanotto@unimi.it (C.Z.); antonia.radaelli@unimi.it (A.R.); 7Veterinary Faculty, University of Ljubljana, Gerbičeva, 60, 1000 Ljubljana, Slovenia; irena.zdovc@vf.uni-lj.si

**Keywords:** gram positive-dependent diseases, antibiotic-resistance, MSSA, MRSA, *Bacillus subtilis*, FtsZ, Z ring, FtsZ inhibition, MIC

## Abstract

Multidrug resistant *Staphylococcus aureus* is a severe threat, responsible for most of the nosocomial infections globally. This resistant strain is associated with a 64% increase in death compared to the antibiotic-susceptible strain. The prokaryotic protein FtsZ and the cell division cycle have been validated as potential targets to exploit in the general battle against antibiotic resistance. Despite the discovery and development of several anti-FtsZ compounds, no FtsZ inhibitors are currently used in therapy. This work further develops benzodioxane-benzamide FtsZ inhibitors. We seek to find more potent compounds using computational studies, with encouraging predicted drug-like profiles. We report the synthesis and the characterization of novel promising derivatives that exhibit very low MICs towards both methicillin-susceptible and -resistant *S. aureus*, as well as another Gram positive species, *Bacillus subtilis,* while possessing good predicted physical-chemical properties in terms of solubility, permeability, and chemical and physical stability. In addition, we demonstrate by fluorescence microscopy that Z ring formation and FtsZ localization are strongly perturbed by our derivatives, thus validating the target.

## 1. Introduction

*Staphylococcus aureus* is a Gram positive, spherical, non-sporulating, non-motile microorganism that grows in characteristic grape-like clusters [1]. As a human commensal, it persistently and asymptomatically colonizes the nares of ~20% of the healthy population [2,3]. *S. aureus* is a highly successful human opportunistic pathogen and is associated with numerous hospital- and community-acquired infections. The main clinical manifestations include skin and soft tissue infections, osteoarticular infections, bacteremia, infective endocarditis, pneumonia, and device-related infections [4]. Its success as a pathogen is largely enabled by its ability to exchange a diverse set of virulence and antibiotic resistance factors through horizontal gene transfer and clonal expansion, allowing for fast evolution and adaptation to new environments and antibiotic therapies [5,6]. Since the early days of antimicrobial chemotherapy, *S. aureus* showed its propensity to rapidly develop resistance to antibiotics. Early reports of penicillinase-producing isolates of *S. aureus* date back to 1944, only a few years after the clinical introduction of penicillin, and ten years later, ~80% of collected isolates showed penicillin resistance [7]. Methicillin-resistant *Staphylococcus aureus* (MRSA) was first reported in 1961 [8], less than one year after the clinical introduction of methicillin, a penicillinase-resistant beta-lactam specifically developed to counteract the growing phenomenon of antibiotic resistance. Over the course of the following decades, resistance to most classes of antimicrobials has been widely reported [4], and currently, most strains of MRSA are considered multidrug resistant. Today, MRSA is a major global healthcare threat and is among the most frequently isolated pathogens in infections in several areas of the world, including Europe, the United States, and East Asia [9]. MRSA is classified as a high priority target for developing novel antibiotics by WHO, ranking highest among Gram positive bacteria (together with vancomycin-resistant *Enterococcus faecium*) [10]. The current standard line of therapy for MRSA infections is represented by vancomycin, daptomycin, and linezolid. While still valid, these options have some limitations, e.g., growing resistance (vancomycin), parenteral administration (vancomycin, daptomycin), or serious adverse effects (linezolid). Given the great adaptability and extensive antibiotic resistance of this microorganism, new molecules based on novel scaffolds, as well as novel targets, are of paramount importance in the context of MRSA infections.

FtsZ, an essential bacterial division protein, has emerged in the last decade as a putative unconventional target, since it plays a crucial role in bacterial replication and viability [11]. Moreover, FtsZ is highly conserved among bacterial species [12], and although it is a functional homolog of human tubulin, their sequences and structures are divergent [13]. Among all the FtsZ inhibitors developed so far, benzamide compounds are the most studied, thanks to their excellent anti-staphylococcal activity, low cytotoxicity, wide chemical accessibility, and the interesting results obtained in association with other antibiotic classes [14,15,16,17]. Along these lines, we recently developed a class of FtsZ inhibitors [18,19,20,21,22] that contain a 2,6-difluoro-benzamide scaffold linked to a differently substituted 1,4-benzodioxane ring. Some of our compounds showed promising MICs for *S. aureus*, almost comparable to the well-known anti-FtsZ benzamides PC190723 and 8j. Their structures, as well those of PC190723 and 8j [23,24], are reported in Figure 1, together with their MICs on MSSA ATCC29213. 

As a continuation of our work on this topic, we considered our latest modeling results, showing how the deepest part of the FtsZ cavity, involved in binding the benzodioxane scaffold, is characterized by narrowness and hydrophobicity [20]. Moreover, docking studies suggested that derivatives presenting an ethylenoxy spacer linker should maintain the same binding mode as compounds with a methylenoxy linker, and attain an even better docking score than their analogs. We recently verified our hypothesis with compounds I and II [22], where increasing the length of the chain retained the ability to interact with FtsZ. Conversely, the propylenoxy bridge of III seemed not to allow a proper fit of the compound.

In this paper, we started with derivatives V–VII, bioisosteres of IV, and with MICs comparable to the precursor IV. In addition, these molecules displayed attractive fitting in the hydrophobic subpocket and good docking scores, while improving their predicted physical chemical properties and metabolic stability by avoiding the ester liability of IV. Based on computational studies, we thus designed and obtained the corresponding superior homologs 1–3, (Figure 2). We then asked whether further lipophilic lengthening on the benzodioxane axis scaffold could allow for a better fit into the FtsZ subpocket and for additional hydrophobic interactions, as encouraged by our computational work. We decided to evaluate naphthalene (4–6), which could be tolerated given the hydrophobic residues present in the environment. Specifically, derivatives 4 and 5 harbor a naphthodioxane along with a methylenoxy or ethylenoxy linker. In contrast, 6 is a simplified derivative in which we eliminated the dioxane ring by removing the O(4), adding extra flexibility to the molecule. Moreover, we also added a tetrahydronaphthodioxane, resulting in compounds 7–8, keeping both the linkers. 

For this work, we focused mainly on the antimicrobial activity of different *S. aureus* strains. We also tested the compounds on the non-pathogen *Bacillus subtilis*, a well-established model for the study of bacterial cell division and activity of FtsZ inhibitors.

## 2. Results

### 2.1. Design and Computational Studies

The chemical structures of compounds **1**–**8** were designed starting from **II** and **V–VII** and considering the chemical features of the FtsZ binding site, in terms of lipophilicity and space. Specifically, we would evaluate an elongation from the reference molecule, either in the linker connecting the 1,4-benzodioxane and the 2,6-difluorobenzamide (**1–3**, **5**, and **8** in Figure 2), or adding hydrophobic substituents on the benzodioxane scaffold, (**4**, **6**, and **7** in Figure 2). Nevertheless, as done before [20,22], we first studied in silico compounds **1–8**, evaluating their docking poses and calculating their physicochemical properties, to assess their attractiveness. 

We previously demonstrated by docking analyses that the binding site of these compounds was the interdomain active site of the protein [20,22]. Moreover, in this work, we aimed at confirming the capability of the 2,6-difluorobenzamide to form three hydrogen bonds with the *S. aureus* FtsZ protein. Specifically, NH_2_ acted as hydrogen bond donor with Val207 and Asn263, while the carbonyl function as a hydrogen bond acceptor with Leu209. Figure 3 presents the docking results for **2**, **3**, **5**, and **8**, the most active compound of this series, as an example.

Considering all the docking poses and scores, no restrictions were set in the docking process, and all the compounds proved to maintain the triple H-bond in the benzamide motif described previously. Moreover, the benzodioxane scaffold, as well as all the substituents, were finely tolerated and were able to be accommodated in the hydrophobic pocket surrounded by the hydrophobic residues Met98, Phe100, Val129, Ile162, Leu190, Gly193, Ile197, Val214, Met218, Met226, Leu261, and Ile311.

Moreover, we evaluated how the linker between the benzamide and the benzodioxane is responsible for the correct fitting of the molecules in the hydrophobic subpocket. Based on this, the hypothesized 2C linker should be a good compromise among the three lengths (1C, 2C, and 3C), conferring adequate but not excessive flexibility, which permits optimal binding of both the aromatic moieties. Indeed, both the above-described H-bonds by the 2,6-difluorobenzamide and the hydrophobic interactions of the 1,4-benzodioxane into the narrow and highly hydrophobic subpocket were retained and maximized.

### 2.2. Physicochemical and Drug-Like Profile Calculations

We chose a wider variety of parameters, to cover all the possible concerns of these interesting derivatives. Thus, we considered an important number of the most relevant molecular and physicochemical properties, as well as some important pharmacokinetics predictions, as summarized in Table 1 [25].

The predicted analyses of the molecules indicate that the majority of the novel proposed FtsZ inhibitors exhibited adequate physicochemical properties and a favorable drug-like profile, meeting the rule of 5 with no outliers. Moreover, focusing on the physicochemical properties, all the compounds showed encouraging permeabilities and no concerns about a potential hERG liability, according to the prediction. Finally, we also considered the Stars parameter, which compares property or descriptor values that fall outside the 95% range of similar values for known drugs, and it was optimal for all the compounds. In detail, the parameters evaluated for #stars were MW, dipole, IP, EA, SASA, FOSA, FISA, PISA, WPSA, PSA, volume, #rotor, donorHB, accptHB, glob, QPpolrz, QPlogPC16, QPlogPoct, QPlogPw, QPlogPo/w, logS, QPLogKhsa, QPlogBB, #metabol. As a result of the promising features of these novel derivatives, which should not have any issue for further therapeutic development, we proceeded to synthesize them and evaluate their antimicrobial profiles.

### 2.3. Chemistry

Scheme 1 shows the synthetic pathway for compounds **1**, **2** and **3**, since the first steps were the same. The synthesis started from the commercially available 3,4-dihydroxybenzonitrile, which was first treated with methyl 3,4-dibromobutyrate, achieving the 7-substituted-1,4-benzodioxane ring, and thus, **9**, with good yields. Only low quantities of the 6-substituted derivative were detected via NMR, when comparing our ^1^H NMR aromatic signals to those of similar 6-substituted benzodioxane derivatives reported in the literature [26], and these traces were easily removed by crystallization of **9** in methanol. After the reduction of the carboxylic group, the alcohol (**10**) was mesylated (**11**) and substituted with 2,6-difluoro-3-hydroxybenzamide (**12**), giving **13** as a white solid. The last mutual intermediate **14** was obtained in quantitative yield by reaction with hydroxylamine hydrochloride in the presence of potassium carbonate. The final compounds **1** and **2** were obtained from **14** by treatment with the proper acetic or propionic anhydride and subsequent treatment with aqueous NaOH, while the achievement of the final compound **3** occurred through 5-mercapto-1,2,4-oxadiazolic ring closure with 1,1′-thiocarbonyldiimidazole (**15**) and subsequent methylation with methyl iodide.

The synthesis of compounds **4–7** and **5–8** were similar, starting from the commercially available naphthalene-2,3-diol or from 5,6,7,8-tetrahydronaphthalene-2,3-diol, obtained by hydrogenation of 3-benzyloxy-5,6,7,8-tetrahydro-2-naphthol [27]. To isolate **4** and **7** (Scheme 2), the naphthalene-2,3-diol or the 5,6,7,8-tetrahydronaphthalene-2,3-diol were treated with epibromohydrin, yielding 2-hydroxymethylnaphtho- or tetrahydronaphthodioxane (**16, 17**). The hydroxylic groups of these intermediates were mesylated (**18, 19**) and substituted with 2,6-difluoro-3-hydroxybenzamide **12,** achieving the final compounds **4** and **7**.

For compounds **5** and **8**, the reaction of naphthodioxane- or tetrahydronaphthodioxane-2,3-diol (Scheme 3) with methyl 3,4-dibromobutyrate gave the compounds **16** and **17**. The reduction with LiAlH_4_ yielded the primary alcohol (**18**, **19**). The final compounds **5** and **8** were obtained first by mesylation (**24, 25**) and subsequent reaction with **12**. Lastly, to isolate compound **6,** we developed a simple synthetic procedure (Scheme 4). The initial step was represented by the treatment of 2-naphthol with 3-chloropropan-1-ol, in basic conditions. This allows the isolation of 3-(2-naphthoxy) propanol (**26**). This intermediate was then subsequently mesylated and substituted with **12**, yielding the desired compound **6**.

### 2.4. Antimicrobial Activity

We tested **1–8** for their antimicrobial activity on different *S. aureus* strains (Table 2). As done before, we considered a methicillin-sensitive *S. aureus* (MSSA, ATCC 29213), a methicillin-resistant *S. aureus* (MRSA, ATCC 43300), and two *S. aureus* strains from the clinic, which showed diverse multidrug resistance. In detail, MDRSA 12.1 shows resistance towards kanamycin, streptomycin, gentamicin, sulfamethoxazole, rifampicin, and tetracycline, while MDRSA 11.7 is resistant to ciprofloxacin, clindamycin, erythromycin, quinupristin, and dalfopristin in combination, tetracycline, tiamulin, and trimethoprim.

We first determined the minimal inhibitory concentration (MIC), i.e., the lowest compound dose (µM) arresting bacterial growth, and the minimal bactericidal concentration (MBC), i.e., the minimal dose (µM) of the compound required for an irreversible block, even after drug removal. Secondly, the derivatives having the most promising activities vs. MRSA were also tested on human MRC-5 cells, calculating their percentages of cytotoxicity by using the MTT assay. Cells were first incubated with each compound for 24 h, then the derivative was removed, and the cells were overlaid with MTT for an additional 3 h. After that time, DMSO replaced the MTT solution, and after 10 min the absorbance was measured at 570 nm. The percentage of cytotoxicity was defined by the formula [100 − (sample OD/untreated cells OD) × 100]. Table 2 reports TD90, defined as the compound concentration (µM) that reduced viability of MRC-5 cells by 90%. Moreover, we reported the therapeutic index (TI), as TD90 and MBC ratio.

All the results of MICs for **1–8**, as well as those for **I**, **II**, and **V–VII** as reference compounds, are presented in Table 2. The data were very promising, with some new compounds exhibiting an order of magnitude lower MIC than the parent compounds. All the compounds had both bacteriostatic and bactericidal properties, and none of them showed concerns in terms of human cytotoxicity.

### 2.5. Effects on B. subtilis

The promising properties of these new compounds prompted us to test them on *B. subtilis*, a model Gram-positive species used previously in evaluating other benzamides, such as **8j**, the benzothiazole derivative of **PC190723** [28]. We chose derivatives **1**, **5**, and **8**, which had the lowest MICs on *S. aureus* and because they were representative of different substituents on the benzodioxane scaffold. *B. subtilis* strain WM5126 was grown until early log phase, and then 1 × and 10 × dilutions of the culture were spotted onto LB plates containing 0, 0.03, 0.06, 0.1, or 1 µg/mL final concentration of the compounds. Compounds **5** and **8** were the most potent against *B. subtilis* with MICs under 0.1 µg/mL, whereas compound **1** had an MIC between 0.1 and 1 µg/mL (Figure 4). These trends were perfectly comparable to what we observed with *S. aureus* MICs.

## 3. Discussion

Predicted properties, antimicrobial assays on both *S. aureus* and *B. subtilis*, and docking poses and scores led us to reach several interesting conclusions.

First, the oxadiazole derivatives **1–3** clearly exhibit improved antimicrobial activity. Specifically, **1** and **2** are 5- and 10-fold more potent on both MRSA and MSSA than their inferior homologs **VI** and **VII**, respectively. A similar outcome is shown with MDRSA, as **1** and **2** possess MICs identical to their activity on MRSA and MSSA. Furthermore, their cytotoxicity on MRC-5 cells revealed no differences when compared to the inferior homologs, thus resulting in a significant improvement in both therapeutic indexes. Even better are the results for compound **3**, which is three- and two-times more active than **V** on methicillin resistant- and sensitive-strains and MDRSA, respectively. Furthermore, the cytotoxicity of **3** is very low, with a consequent high and desirable therapeutic index. In addition, if we compare MICs and MBCs, it is noticeable that compounds **1–3** exhibit no differences between bacteriostatic and bactericidal potencies. Computational studies suggest that these three differently substituted oxadiazoles positively drive the benzodioxane scaffold into the hydrophobic subpocket, enhancing their interaction with FtsZ.

The lipophilic and spatial features of the FtsZ binding cavity perfectly explain the differences in antimicrobial properties of compounds **4/5** and **7/8**. They all show higher antimicrobial activities than non-substituted benzodioxanes **I** and **II**, and the relative differences between methylenoxy and ehylenoxy derivatives are maintained, with superior homologs 12- (**5**) and 50- (**8**) fold more potent than inferior ones (**4** and **7**). There are no differences between MICs and MBCs, and the activity on MDRSA is maintained. Compound **8** ended up being the best of this series, both for MICs and cytotoxicity, with a consequent outstanding TI. Its potency is likely related to the hydrophobic interactions generated by the tetrahydronaphthalene in the binding subpocket, which is characterized by hydrophobic and non-aromatic residues as described in the computational section.

Additionally, comparing **5** and **6**, no improvements were achieved by simplifying the structure and avoiding the dioxane moiety. This emphasizes the importance of the benzodioxane ring for permitting the correct fitting of the molecules into the hydrophobic subpocket.

Finally, we decided to evaluate the best derivatives on *B. subtilis* FtsZ rings by fluorescence microscopy, to validate FtsZ as the target of these compounds. Indeed, we previously used morphometric analysis [20,22], as well as in vitro biochemical assays [21], including a GTPase activity assay and a polymerization activity assay to demonstrate the typical alterations of cell division and FtsZ inhibition.

Here we took advantage of the large size of *B. subtilis* cells and strains containing fluorescent proteins that localize to the FtsZ ring to assess the effects of compounds **7** and **8** on *B. subtilis* FtsZ rings by fluorescence microscopy. Direct effects on FtsZ ring localization in cells would further validate FtsZ as the target of this class of compounds, by analogy to previous investigations of compound 8j [28].

In the absence of our derivatives, all cells displayed normal sharp bands of GFP-ZapA, a well-established proxy for FtsZ and the Z ring [29] (yellow arrows in Figure 5 highlight normal sharp bands at the future division sits). On the contrary, in cells treated with compounds **7** and **8**, we observed GFP-ZapA forming many (0.4 µg/mL **7**) or nearly all (0.8 µg/mL **7** or 0.4 µg/mL **8**) foci or focal clusters (white arrows in Figure 5), and abnormal cell elongation is clearly evident. The perturbation of Z rings and cell elongation is similar to the previously described effects of **8j**. The weaker effect of **7** on Z rings vs. **8** is consistent with the significantly higher MIC of **7** on *S. aureus* (Table 2). These results are clearly consistent with the Z ring and FtsZ as targets of this class of compounds.

## 4. Materials and Methods

### 4.1. Chemistry

All the reagents and the solvents were used without purification or distillation, after purchasing from commercial sources (Merck, Fluorochem, and TCI).

Silica gel matrix, having fluorescent indicator 254 nm, was used both in TLC (thin-layer chromatography, on aluminum foils), and in flash chromatography (particle size 40–63 µm, Merck) on Puriflash XS 420 (Sepachrom Srl, Rho (MI), Italy). The visualization was with UV light at 254 nm (λ).

Varian (Palo Alto, CA, USA) Mercury 300 NMR spectrometer/Oxford Narrow Bore superconducting magnet operating at 300 MHz was used for all ^1^H-NMR spectra. ^13^C-NMR spectra were acquired at 75 MHz. We reported all chemical shifts (δ) in ppm, relative to residual solvent as internal standard. The following abbreviations refer to signal multiplicity: s = singlet, d = doublet, dd = doublet of doublets, t = triplet, q = quadruplet, dq = doublet of quadruplets, m = multiplet, bs = broad singlet.

The final products, **1–8,** were analyzed by reverse-phase HPLC using a Waters XBridge C-18 column (5 μm, 4.6 mm × 150 mm) on an Elite LaChrom HPLC system with a diode array detector (Hitachi, San Jose, CA; USA). Mobile phase: A, H_2_O with 0.10% TFA; B, acetonitrile with 0.10% TFA; gradient, 90% A to 10% A in 25 min with 35 min run time and a flow rate of 1 mL/min. Their purity was quantified at peculiar λ max values, depending on the compound, and all resulted in >95%. The relative retention times are reported in each experimental section. Melting points were determined by DSC analysis using a DSC 1020 apparatus (TA Instruments, New Castle, DE, USA).

The ^1^H- and ^13^C-NMR spectra of compounds **1**-**8**, together with their HPLC profiles, are included in the Appendix A.

#### Synthesis

**Methyl (7-cyano-1,4-benzodioxan-2-yl)-acetate (9):** A solution of 3,4-dihydroxybenzonitrile (2 g, 14.80 mmol) in acetone (20 mL) was added of potassium carbonate (4.91 g, 35.52 mmol). The reaction mixture was kept stirring at room temperature for 30 min, then methyl 3,4-dibromobutyrate (4.23 g, 16.68 mmol) was added dropwise, and the medium was heated at reflux. The reaction mixture was then stirred at that temperature for 3 h, letting the completion of the reaction. After concentration under vacuum, the crude was diluted with ethyl acetate (50 mL), washed with 10% aqueous NaOH, and 10% aqueous NaCl (2 × 20 mL), dried over Na_2_SO_4_, filtered, and concentrated to give a residue. Crystallization from methanol (3 vol) gave 1.40 g of **9** as a white solid. M.p. 143.16 °C Yield: 41% **^1^H NMR (300 MHz, CDCl_3_, δ):** 7.14 (m, 2H), 6.92 (d, *J* = 9.0 Hz, 1H), 4.62 (dq, *J* = 6.6, 2.2 Hz, 1H), 4.39 (dd, *J* = 11.5, 2.2 Hz, 1H), 4.04 (dd, *J* = 11.5, 6.6 Hz, 1H), 3.75 (s, 3H), 2.78 (dd, *J* = 16.3, 6.6 Hz, 1H), 2.64 ppm (dd, *J* = 16.3, 6.6 Hz, 1H).

**2-Hydroxyethyl-7-cyano-1,4-benzodioxane (10):** A solution of **9** (1.40 g, 6.00 mmol) in dry THF (10 mL) was added dropwise to a suspension of LiAlH_4_ (0.21 g, 6.00 mmol) in dry THF (5 mL) at –40 °C under N_2_ atmosphere. The reaction mixture was stirred at –35 °C for 30 min, then diluted with ethyl acetate (15 mL), washed with 10% aqueous HCl, water and 10% aqueous NaCl (3 × 10 mL), dried over Na_2_SO_4_, filtered, and concentrated to give 0.95 g of **10** as a yellowish oil. Yield: 82% **^1^H NMR (300 MHz, CDCl_3_, δ):** 7.13 (m, 2H), 6.92 (m, 1H), 4.37 (m, 2H), 3.95 (m, 3H), 1.94 ppm (m, 2H).

**2-methansulfonyloxyethyl-7-cyano-1,4-benzodioxane (11): 10** (0.95 g, 4.63 mmol) was dissolved in DCM (10 mL) and TEA (0.97 mL, 6.94 mmol), then added of mesyl chloride (0.54 mL, 6.94 mmol), dropwise, at 0 °C. The reaction mixture was stirred at room temperature for 3.5 h, till reaction completion. Then diluted with DCM (15 mL), washed firstly with 10% aqueous NaHCO_3_ (5 mL), secondly with 10% aqueous HCl (5 mL) and finally with 10% aqueous NaCl (10 mL), filtered, and concentrated under vacuum to give 1.18 g of **11** as a yellowish oil. Yield: 90% **^1^H NMR (300 MHz, CDCl_3_, δ):** 7.17 (m, 2H), 6.93 (d, *J* = 8.9 Hz, 1H), 4.47 (m, 2H), 4.35 (m, 2H), 3.99 (dd, *J* = 11.6, 7.5 Hz, 1H), 3.05 (s, *J* = 1.9 Hz, 3H), 2.09 ppm (m, 2H).

**3-[2-(7-cyano-1,4-benzodioxan-2-yl)ethyloxy]-2,6-difluorobenzamide (13)**: A solution of **12** (0.76 g, 4.37 mmol) in dry DMF (5 mL) under N_2_ atmosphere was amounted of potassium carbonate (0.63 g, 4.58 mmol). After stirring at room temperature for 30 min, a solution of **11** (1.18 g, 4.16 mmol) in dry DMF (5 mL) was added. The reaction mixture was stirred at 80 °C for 4 h, till reaction completion, and then concentrated under vacuum, diluted with ethyl acetate (15 mL), washed with 10% aqueous NaCl (4 × 10 mL), dried over Na_2_SO_4_, filtered, and concentrated to give a residue which was purified by flash chromatography. Elution with 55/45 cyclohexane/ethyl acetate gave 1.02 g of **13** as a white solid. Yield: 68% M.p. 172.1 °C **^1^H NMR (300 MHz, CDCl_3_, δ):** 7.15 (m, 2H), 7.02 (m, 1H), 6.89 (m, 2H), 6.21 (bs, 1H), 6.05 (bs, 1H), 4.45 (m, 2H), 4.25 (m, 2H), 4.05 (m, 1H), 2.16 ppm (m, 2H).

**3-[2-(7-N’-hydroxycarbamimidoyl-1,4-benzodioxan-2-yl)ethyloxy]-2,6-difluorobenzamide (14):** A solution of **12** (1.00 g, 2.77 mmol) and hydroxylamine hydrochloride (0.96 g, 13.87 mmol) in dry DMF (10 mL) was added of a solution of potassium carbonate (1.91 g, 13.87 mmol) in water (5 mL). The reaction mixture was stirred at 80 °C for 18 h, concentrated under vacuum, diluted with ethyl acetate (15 mL), washed with 10% aqueous NaCl (4 × 10 mL), dried over Na_2_SO_4_, filtered, and concentrated to give a 1.01 g of **14** as a yellowish oil. Yield: 95% **^1^H NMR (300 MHz, CD_3_OD, δ):** 7.21 (td, *J* = 9.2, 5.2 Hz, 1H), 7.12 (m, 2H), 6.96 (td, *J* = 2.3, 0.7 Hz, 1H), 6.85 (d, *J* = 8.4 Hz, 1H), 4.39 (m, 2H), 4.27 (m, 2H), 4.00 (dd, *J* = 11.8, 7.7 Hz, 1H), 2.14 ppm (m, 2H).

**3-[2-(7-(5-methyl-1,2,4-oxadiazol-3-yl)-1,4-benzodioxan-2-yl)ethyloxy]-2,6-difluorobenzamide (1):** Acetic anhydride (0.07 mL, 0.76 mmol) was added to a solution of **14** (0.25 g, 0.63 mmol) and pyridine (0.07 g, 0.95 mmol) in dry DMF and CHCl_3_ (10 mL + 2 mL). After stirring at reflux for 2 h, 2.5 N aqueous NaOH (1 mL) was added and the reaction was and stirred for 18 h. The reaction mixture was then concentrated under vacuum, diluted with ethyl acetate (15 mL), washed with 10% aqueous NaCl (4 × 10 mL), dried over Na_2_SO_4_, filtered, and concentrated to give a residue. Digestion with methanol (20 vol.) gave 0.09 g of **1** as a white solid. Yield: 34% M.p. 164.4 °C, Tr (HPLC, Appendix A): 12.9 min, Purity = 95.5%. **^1^H NMR (Appendix A, 300 MHz, d_6_-DMSO, δ):** 8.09 (s, 1H), 7.82 (s, 1H), 7.45 (m, 2H), 7.27 (td, *J* = 9.3, 5.3 Hz, 1H), 7.05 (m, 2H), 4.43 (m, 2H), 4.26 (m, 2H), 4.07 (dd, *J* = 11.4, 7.1 Hz, 1H), 2.61 (s, 3H), 2.09 ppm (m, 2H). **^13^C NMR (Appendix A, 75 MHz, d_6_-DMSO, δ):** 176.56, 167.54, 161.73, 152.40 (dd, *J* = 240.0, 6.8 Hz), 148.40 (dd, *J* = 247.1, 8.6 Hz), 146.1, 143.5, 143.30 (dd, *J* = 10.9, 3.4 Hz), 120.81, 119.96, 118.15, 117.10 (dd, *J* = 24.7, 20.2 Hz), 116.70, 116.07, 111,40 (dd, *J* = 22.5, 3.8 Hz) 70.74, 67.87, 65.81, 20.32, 12.41 ppm.

**3-[2-(7-(5-ethyl-1,2,4-oxadiazol-3-yl)-1,4-benzodioxan-2-yl)ethyloxy-2,6-difluorobenzamide (2):** Propionic anhydride (0.10 mL, 0.76 mmol) was added to a solution of **14** (0.25 g, 0.63 mmol) and pyridine (0.07 g, 0.95 mmol) in dry DMF and CHCl_3_ (10 mL + 2 mL). After stirring at reflux for 2 h, 2.5 N aqueous NaOH (1 mL) was added and the reaction was and stirred for 18 h. The reaction mixture was then concentrated under vacuum, diluted with Ethyl acetate (15 mL), washed with 10% aqueous NaCl (4 × 10 mL), dried over Na_2_SO_4_, filtered, and concentrated to give a residue which was purified by flash Chromatography. Elution with 1/1 cyclohexane/ethyl acetate and subsequent digestion with methanol (20 vol.) gave 0.02 g of **2** as a white solid. Yield: 7% M.p. 151.1 °C, Tr (HPLC, Appendix A): 9.9 min, Purity = 99.4%.**^1^H NMR (Appendix A, 300 MHz, d_6_-DMSO, δ):** 8.09 (s, 1H), 7.82 (s, 1H), 7.46 (m, 2H), 7.28 (td, *J* = 9.3, 5.3 Hz, 1H), 7.06 (td, *J* = 9.1, 1.9 Hz, 1H), 7.03 (d, *J* = 8.4 Hz, 1H), 4.44 (m, 2H), 4.27 (m, 2H), 4.07 (dd, *J* = 11.4, 7.1 Hz, 1H), 2.96 (q, *J* = 7.6 Hz, 2H), 2.11 (m, 2H), 1.30 ppm (t, *J* = 7.6 Hz, 3H). **^13^C NMR (Appendix A, 75 MHz, d_6_-DMSO, δ):** 181.37, 167.40, 161.72, 152.30 (dd, *J* = 239.6, 7.1 Hz), 148.40 (dd, *J* = 246.8, 8.2 Hz), 146.10, 143.49, 143.30 (dd, *J* = 10.5, 3.0 Hz), 129.73, 120.84, 120.00, 118.16, 117.10 (dd, *J* = 24.7, 20.2 Hz), 116.08, 116.00, 111,40 (dd, *J* = 22.5, 3.8 Hz) 70.73, 67.88, 65.80, 30.32, 19.98, 10.88 ppm.

**3-[2-(7-(5-mercapto-1,2,4-oxadiazol-3-yl)-1,4-benzodioxan-2-yl)ethyloxy]-2,6-difluorobenzamide (15):** 1,1′-Thiocarbonyldiimidazol (0.34 g, 1.91 mmol) was added to a solution of **14** (0.5 g, 1.27 mmol) and DBU (0.76 mL, 5.08 mmol) in dioxane (10 mL) under N_2_ atmosphere. The reaction mixture was stirred at room temperature for 3.5 h, then concentrated under vacuum, diluted with ethyl acetate and 10% aqueous HCl (2 × 15 mL), washed with 10% aqueous NaCl (3 × 10 mL), dried over Na_2_SO_4_, filtered, and concentrated to give 0.47 g of **15** as a yellow oil. Yield: 85% **^1^H NMR (300 MHz, CD_3_OD, δ):** 7.35 (m, 2H), 7.22 (m, 1H), 6.96 (m, 1H), 4.45 (m, 2H), 4.29 (m, 2H), 4.06 (dd, *J* = 11.9, 7.9 Hz, 1H), 2.11 ppm (m, 2H).

**3-[2-(7-(5-methylthio-1,2,4-oxadiazol-3-yl)-1,4-benzodioxan-2-yl)ethyloxy]-2,6-difluorobenzamide (3):** Potassium carbonate (0.17 g, 1.27 mmol) was added to a solution of **14** (0.46 g, 1.06 mmol) in ACN/DMF (10 + 2 mL) under N_2_ atmosphere. After stirring at room temperature for 30 min, methyl iodide (0.08 mL, 1.27 mmol) was added dropwise. The reaction mixture was stirred at 50 °C for 1.5 h, then concentrated under vacuum, diluted with ethyl acetate and 10% aqueous NaCl (2 × 15 mL), washed with 10% aqueous NaHCO_3_ and 10% aqueous NaCl (2 × 10 mL), dried over Na_2_SO_4_, filtered, and concentrated to give a residue. Digestion with methanol (20 vol.) gave 0.18 g of **3** as a yellowish solid. Yield: 38% M.p. 184.2 °C, Tr (HPLC, Appendix A): 15.0 min, Purity = 95.0%. **^1^H NMR (Appendix A, 300 MHz, d_6_-DMSO, δ):** 8.08 (s, 1H), 7.80 (s, 1H), 7.44 (m, 2H), 7.27 (td, *J* = 9.3, 5.3 Hz, 1H), 7.06 (m, 2H), 4.45 (m, 2H), 4.26 (m, 2H), 4.07 (dd, *J* = 11.4, 7.2 Hz, 1H), 2.78 (s, 3H), 2.11 ppm (m, 2H). **^13^C NMR (Appendix A, 75 MHz, d_6_-DMSO, δ):** 178.90, 167.87, 161.72, 152.40 (dd, *J* = 240.0, 6.8 Hz), 148.40 (dd, *J* = 247.1, 8.6 Hz), 146.34, 143.52, 143.30 (dd, *J* = 10.5, 3.0 Hz), 120.97, 119.35, 118.19, 117.10 (dd, *J* = 24.7, 20.2 Hz), 116.23, 116.10 (dd, *J* = 9.7, 1.5 Hz), 111,40 (dd, *J* = 22.5, 3.8 Hz) 70.75, 67.90, 65.81, 30.33, 15.08 ppm.

**2-hydroxymethyl-2,3-dihydronaphtho[2,3-*b*][1,4]dioxine (16):** Potassium carbonate (1.9 g, 13.73 mmol) was added to a solution of naphthalen-2,3-diol (1 g, 6.24 mmol) in acetone (10 mL). After stirring for 30 min, epibromohydrin (0.59 mL, 6.9 mmol) was added dropwise. The reaction mixture was stirred at RT for 72 h, concentrated under vacuum, diluted with ethyl acetate (35 mL), washed with 10% aqueous NaOH (15 mL) and 10% aqueous NaCl (15 mL), dried with Na_2_SO_4_, filtered, and concentrated under vacuum to yield 1.01 g of **16** as a viscous oil. Yield: 75% **^1^H NMR (300 MHz, CDCl_3_ δ):** 7.66 (m, 2H), 7.30 (m, 4H), 4.37 (m, 2H), 4.15 (m, 1H), 3.91 ppm (m, 2H).

**2-mesyloxymethyl-1,4-naphthodioxane (18):** Prepared from **16** as described for **11** using Mesyl chloride (1.2 eq.) and TEA (1.2 eq) in DCM (10 mL) for 3 h giving **18** as a yellow oil. Yield: 98% **^1^H NMR (300 MHz, CDCl_3_ δ):** 7.66 (m, 2H), 7.32 (m, 4H), 4.59 (m, 1H), 4.49 (d, *J* = 5.3 Hz, 2H), 4.40 (dd, *J* = 11.7, 2.4 Hz, 1H), 4.22 (dd, *J* = 11.7, 6.6 Hz, 1H), 3.11 ppm (s, 3H).

**3-[(2,3-dihydronaphtho[2,3-*b*][1,4]dioxin-2-yl)methoxy]-2,6-difluorobenzamide (4):** Prepared from **18** as described for **13** using Potassium carbonate (1.1 eq) and 2,6-difluoro-3-hydroxybenzamide (1.05 eq) in dry DMF (5 mL) at 60 °C for 24 h and purified by flash chromatography on silica gel. Elution with 1/1 Cyclohexane/Ethyl acetate gave 0.20 g of **4** as a white solid. Yield: 40% M.p. 157.4 °C, Tr (HPLC, Appendix A): 14.5 min, Purity = 97.0%. **^1^H NMR (Appendix A, 300 MHz, d_6_-DMSO, δ):** 8.13 (bs, 1H), 7.86 (bs, 1H), 7.53 (m, 2H), 7.30 (m, 5H), 7.07 (td, *J* = 9.0, 1.9 Hz, 1H), 4.73 (m, 1H), 4.51 (dd, *J* = 11.6, 2.5 Hz, 1H), 4.39 (dd, *J* = 11.4, 4.6 Hz, 1H), 4.34. (dd, *J* = 11.4, 5.8 Hz, 1H), 4.24 ppm (dd, *J* = 11.6, 7.3 Hz, 1H). **^13^C NMR (Appendix A, 75 MHz, d_6_-DMSO, δ):** 161.66, 152.60 (dd, *J* = 240.0, 6.8 Hz), 148.40 (dd, *J* = 247.1, 8.6 Hz), 148.68, 143.47, 143.20 (dd, *J* = 10.9, 3.4 Hz), 129.66, 129.49, 126.69, 124.63, 117.10 (dd, *J* = 25.1, 20.6 Hz), 116.40 (dd, *J* = 9.0, 2.3 Hz), 112.72, 112.58, 111.50 (dd, *J* = 22.5, 3.7 Hz) 71.98, 68.80, 64.99 ppm.

**Methyl (2,3-dihydronaphtho[2,3-*b*][1,4]dioxin-2-yl)acetate (20):** Prepared from naphthalen-2,3,-diol as described for **9** using Potassium carbonate (2.2 eq.) and methyl 3,4-dibromobutyrate (1.1 eq) for 72 h giving **20** as a dense yellowish oil. Yield: 94% **^1^H NMR (300 MHz, CDCl_3_ δ):** 7.64 (m, 2H), 7.29 (m, 4H), 4.73 (m, 1H), 4.41 (dd, *J* = 11.4, 2.2 Hz, 1H), 4.09 (dd, *J* = 11.4, 6.9 Hz, 1H), 3.76 (s, 3H), 2.85 (dd, *J* = 16.1, 6.8 Hz, 1H), 2.70 ppm (dd, *J* = 16.1, 6.5 Hz, 1H).

**2-(2-hydroxyethyl)-2,3-dihydronaphtho[2,3-*b*][1,4]dioxine (22):** Prepared from **20** as described for **10** using LiAlH_4_ (1.1 eq.) at 0 °C giving **22** as a pale oil. Yield: 85% **^1^H NMR (300 MHz, CDCl_3_ δ):** 7.63 (m, 2H), 7.28 (m, 4H), 4.47 (m, 1H), 4.35 (dd, *J* = 11.4, 2.3 Hz, 1H), 4.05 (dd, *J* = 11.4, 7.8 Hz, 1H), 3.95 (m, 2H), 1.92 ppm (m, 2H).

**2-(2-mesyloxyethyl)-2,3-dihydronaphtho[2,3-*b*][1,4]dioxine (24):** Prepared from **22** as described for **11** using mesyl chloride (1.2 eq.) and TEA (1.2 eq) in DCM (10 mL) for 3 h giving **24** as a yellow oil. Yield: 93% **^1^H NMR (300 MHz, CDCl_3_ δ):** 7.65 (m, 2H), 7.30 (m, 4H), 4.51 (m, 3H), 4.36 (dd, *J* = 11.4, 2.3 Hz, 1H), 4.06 (dd, *J* = 11.4, 7.2 Hz, 1H), 3.06 (s, 3H), 2.14 ppm (m, 2H).

**3-[2-(2,3-dihydronaphtho[2,3-*b*][1,4]dioxin-2-yl)ethoxy]-2,6-difluorobenzamide (5):** Prepared from **24** as described for **13** using potassium carbonate (1.1 eq) and **12** (1.05 eq) in dry DMF (5 mL) at 60 °C for 24 h and purified by flash chromatography on silica gel. Elution with 1/1 cyclohexane/ethyl acetate gave 0.12 g of **5** as a white solid. Yield: 23% M.p. 140.9 °C, Tr (HPLC, Appendix A): 15.2 min, Purity = 99.3%. **^1^H NMR (Appendix A, 300 MHz, d_6_-DMSO, δ):** 8.11 (bs, 1H), 7.83 (bs, 1H), 7.68 (m, 2H), 7.33 (d *J* = 1.9Hz, 1H), 7.31–7.22 (m, 4H), 7.06 (td, *J* = 9.0, 1.9 Hz, 1H), 4.47 (m, 2H), 4.27 (t, *J* = 6.2 Hz, 2H), 4.09 (dd, *J* = 12.0, 7.9 Hz, 1H), 2.12 ppm (m, 2H). **^13^C NMR (Appendix A, 75 MHz, d_6_-DMSO, δ):** 161.76, 152.35 (dd, *J* = 238.8, 6.8 Hz), 148.39 (dd, *J* = 246.9, 8.4 Hz), 143.84, 143.74, 143.31 (dd, *J* = 10.9, 3.1 Hz), 129.57, 129.49, 126.64, 124.53, 117.07 (dd, *J* = 24.9, 20.4 Hz), 116.00 (dd, *J* = 9.3, 2.3 Hz), 112.65, 112.34, 111.41 (dd, *J* = 22.7, 4.0 Hz), 70.82, 67.73, 65.86, 30.49 ppm.

**3-(naphthalen-2-yl)propanol (26):** Potassium carbonate (5.75 g, 41.62 mmol) was added to a solution of 2-naphthol (3 g, 20.81 mmol), and potassium iodide (0.34 g, 2.08 mmol) in dry DMF (30 mL). After stirring at room temperature for 30 min, 3-chloro-1-propanol (1.91 mL, 22.89 mmol) was added dropwise. The reaction mixture was stirred at 70 °C for 24 h, concentrated under vacuum, diluted with ethyl acetate (50 mL), washed with 10% aqueous NaOH and 10% aqueous NaCl (2 × 20 mL), dried over Na_2_SO_4_, filtered, and concentrated to give 4.21 g of **26** as a white solid. Yield: Quantitative, M.p. 99.0 °C (lit.) **^1^H NMR (300 MHz, CDCl_3_ δ):** 7.74 (m, 3H), 7.44 (t, *J* = 7.5 Hz, 1H), 7.33 (t, *J* = 8.0 Hz, 1H), 7.14 (m, 2H), 4.26 (t, *J* = 6.0 Hz, 2H), 3.92 (t, *J* = 6.0 Hz, 2H), 2.12 ppm (p, *J* = 6.0 Hz, 2H).

**3-(naphthalen-2-yl propyl-1-methansulfonate (27):** Prepared from **26** as described for **11** using mesyl chloride (1.2 eq.) and TEA (1.2 eq) in DCM (10 mL) for 1.5 h giving **27** as a yellow oil. Yield: Quantitative. **^1^H NMR (300 MHz, CDCl_3_ δ):** δ 7.75 (m, 3H), 7.45 (t, *J* = 7.5 Hz, 1H), 7.35 (m, 1H), 7.13 (m, 2H), 4.50 (t, *J* = 6.0 Hz, 2H), 4.22 (t, *J* = 6.0 Hz, 2H), 2.99 (s, 3H), 2.30 ppm (p, *J* = 6.0 Hz, 2H).

**3-[(3-(naphthalen-2-yl)propyl-1-oxy]-2,6-difluorobenzamide (6):** Prepared from **27** as described for **13** using potassium carbonate (1.1 eq) and **12** (1.05 eq) in dry DMF (5 mL) at 60 °C for 24 h and purified by flash chromatography on silica gel. Elution with 6/4 cyclohexane/ethyl acetate and subsequent crystallization with chloroform (5 vol) gave 0.20 g of **6** as a white solid. Yield: 34% M.p. 120.9 °C, Tr (HPLC, Appendix A): 14.8 min, Purity = 96.4%. **^1^H NMR (Appendix A, 300 MHz, d_6_-DMSO, δ):** 8.08 (s, 1H), 7.80 (m, 4H), 7.43 (m, 1H), 7.31 (m, 2H), 7.24 (dd, *J* = 9.4, 5.3 Hz, 1H), 7.16 (dd, *J* = 9.0, 2.5 Hz, 1H), 7.04 (td, *J* = 9.0, 1.9 Hz, 1H), 4.24 (t, *J* = 6.2 Hz, 4H), 2.24 ppm (p, *J* = 6.2 Hz, 2H). **^13^C NMR (Appendix A, 75 MHz, d_6_-DMSO, δ):** δ 161.76, 156.78, 152.20 (dd, *J* = 239.3, 6.8 Hz), 148.39 (dd, *J* = 246.7, 8.3 Hz), 143.40 (dd, *J* = 11.3, 3.0 Hz), 134.72, 129.76, 128.93, 127.93, 127.13, 126.83, 124.03, 119.12, 117.10 (dd, *J* = 25.1, 20.6 Hz), 115.90 (dd, *J* = 9.0, 2.2 Hz), 111,40 (dd, *J* = 22.9, 4.1 Hz) 107.15, 66.74, 65.55, 28.98 ppm.

**2-hydroxymethyl-2,3,6,7,8,9-hexahydronaphtho[2,3-*b*][1,4]dioxine (17):** Prepared from 5,6,7,8-tetrahydronaphthalene-2,3-diol as described for **16** using potassium carbonate (2.2 eq.) and epibromohydrin (2 eq.) for 18 h and purified by flash chromatography on silica gel. Elution with 6/4 cyclohexane/ethyl acetate gave 0.12 g of **17** as a colourless oil. Yield: 13% **^1^H NMR (300 MHz, CDCl_3_ δ):** 6.61 (s, 1H), 6.59 (s, 1H) 4.22 (m, 2H), 4.05 (m, 1H), 3.85 (m, 2H), 2.65 (m, 4H), 1.77 ppm (m, 4H).

**2-Mesyloxymethyl-2,3,6,7,8,9-hexahydronaphtho[2,3-*b*][1,4]dioxine (19):** Prepared from **17** as described for **11** using mesyl chloride (1.2 eq.) and TEA (1.2 eq) in DCM (10 mL) for 3 h giving **19** as a yellow oil. Yield: 93% **^1^H NMR (300 MHz, CDCl_3_ δ):** 6.59 (s, 2H), 4.42 (m, 1H), 4,42 (m, 2H) 4.26 (dd, *J* = 11.6, 2.0 Hz, 1H), 4.09 (dd, *J* = 11.6, 6.0 Hz, 1H), 3.08 (s, 3H), 2.68 (m, 4H), 1.75 ppm (m, 4H).

**3-(2,3,6,7,8,9-hexahydronaphtho[2,3-*b*][1,4]dioxin-2-yl)methoxy]-2,6-difluorobenzamide (7):** Prepared from **19** as described for **13** using potassium carbonate (1.1 eq) and **12** (1.05 eq) in dry DMF (5 mL) at 80 °C for 4 h and purified by flash chromatography on silica gel. Elution with 1/1 cyclohexane/ethyl acetate and subsequent treatment with DCM lets the precipitation of 0.04 g of **7** as a white solid. Yield: 22% M.p. 156.6 °C, Tr (HPLC, Appendix A): 15.3 min, Purity = 95.1%. **^1^H NMR (Appendix A, 300 MHz, d_6_-DMSO, δ):** 8.12 (s, 1H), 7.84 (s, 1H), 7.28 (td, *J* = 9.1, 5.3 Hz, 1H), 7.06 (td, *J* = 9.1, 1.8 Hz, 1H), 6.55 (s, 2H), 4.52 (m, 1H), 4.28 (m, 2H), 4.27 (m, 1H) 4.08 (dd, *J* = 11.5, 6.8 Hz, 1H), 2.58 (s, 4H), 1.65 ppm (s, 4H). **^13^C NMR (Appendix A, 75 MHz, d_6_-DMSO, δ):** 161.7, 152.5 (dd, *J* = 240.0, 6.8 Hz), 148.4 (dd, *J* = 247.1, 8.6 Hz), 143.2 (dd, *J* = 10.9, 3.4 Hz), 141.0, 140.6, 130.1, 129.8, 117.1 (dd, *J* = 24.8, 20.3 Hz), 117.1, 116.9, 116.3 (dd, *J* = 9.4, 1.9 Hz), 111.4 (dd, *J* = 22.9, 4.1 Hz), 71.7, 68.7, 64.8, 28.5, 23.3 ppm.

**Methyl-(2,3,6,7,8,9-hexahydronaphtho[2,3-*b*][1,4]dioxin2-yl)acetate (21):** Prepared from naphthalen-2,3,-diol as described for **9** using potassium carbonate (2.2 eq.) and methyl 3,4-dibromobutyrate (1.1 eq) for 18 h and purified by flash chromatography on silica gel. Elution with 9/1 cyclohexane/ethyl acetate gave 0.85 g of **21** as a colourless oil. Yield: 52% **^1^H NMR (300 MHz, CDCl_3_ δ):** 6.58 (s, 2H), 4.59 (qd, *J* = 6.6, 2.3 Hz, 1H), 4.26 (dd, *J* = 11.3, 2.3 Hz, 1H), 3.96 (dd, *J* = 11.3, 6.6 Hz, 1H), 3.74 (s, 3H), 2.77 (dd, *J* = 16.1, 6.6 Hz, 1H), 2.64 (m, 4H), 2.63 (dd, *J* = 16.1, 6.6 Hz, 1H), 1.73 ppm (m, 4H).

**2-(2-Hydroxyethyl)-(2,3,6,7,8,9-hexahydronaphtho[2,3-*b*][1,4]dioxine) (23):** Prepared from **21** as described for **10** using LiAlH_4_ (1.0 eq.) at 0 °C giving **23** as a yellowish oil. Yield: 78% **^1^H NMR (300 MHz, CDCl_3_ δ):** 6.57 (s, 2H), 4.33 (m, 1H), 4.21 (dd, *J* = 11.3, 2.2 Hz, 1H), 3.91 (m, 3H) 2.64 (m, 4H), 1.89 (m, 2H), 1.72 ppm (m, 4H).

**2-(2-Mesyloxyethyl)-(2,3,6,7,8,9-hexahydronaphtho[2,3-*b*][1,4]dioxine) (25):** Prepared from **23** as described for **11** using mesyl chloride (1.2 eq.) and TEA (1.2 eq) in DCM (10 mL) for 3 h giving **25** as a white wax. Yield: 87%. **^1^H NMR (300 MHz, CDCl_3_ δ):** 6.57 (s, 2H), 4.46 (m, 2H), 4.31 (ddd, *J* = 13.4, 6.8, 2.2 Hz, 1H), 4.22 (dd, *J* = 11.3, 2.2 Hz, 1H), 3.91 (dd, *J* = 11.3, 6.8 Hz, 1H), 3.04 (s, 3H), 2.65 (m, 4H), 2.04 (m, 2H), 1.75 ppm (m, 4H).

**3-[2-(2,3,6,7,8,9-hexahydronaphtho[2,3-*b*][1,4]dioxin-2-yl)ethoxy]-2,6-difluorobenzamide (8):** Prepared from **25** as described for **13** using potassium carbonate (1.1 eq) and **12** (1.05 eq) in dry DMF (5 mL) at 80 °C for 4 h. Crystallization from IPA (20 vol.) gave 0.24 g of **8** as a white solid. Yield: 50% M.p. 166.5 °C, Tr (HPLC, Appendix A): 16.7 min, Purity = 95.0%. **^1^H NMR (Appendix A, 300 MHz, d_6_-DMSO, δ):** 8.03 (s, 1H), 7.75 (s, 1H), 7.17 (tt, *J* = 9.0, 4.5 Hz, 1H), 6.98 (t, *J* = 9.0 Hz, 1H), 6.44 (s, 2H), 4.17 (m, 4H), 3.82 (m, 1H), 2.48 (m, 4H), 1.96 (m, 2H), 1.56 ppm (m, 4H). **^13^C NMR (Appendix A, 75 MHz, d_6_-DMSO, δ):** 161.7, 152.2 (dd, *J* = 240.0, 6.7 Hz), 148.3 (dd, *J* = 246.7, 9.0 Hz), 143.3 (dd, *J* = 10.5, 3.0 Hz), 141.0, 140.8, 129.7, 129.6, 117.0 (dd, *J* = 25.4, 20.7 Hz), 117.0, 116.7, 115.9 (dd, *J* = 9.4, 1.9 Hz), 111.3 (dd, *J* = 22.5, 4.5 Hz), 70.3, 67.5, 65.8, 30.3, 28.5, 23.2 ppm.

### 4.2. Cells

Normal human lung fibroblasts (MRC-5), as well as the Gram-positive *S. aureus* (methicillin-sensitive, MSSA ATCC 29213, and methicillin-resistant, MRSA ATCC 43300) were grown in Dulbecco’s modified Eagle’s medium (DMEM) or in Luria-Bertani broth (LB), as reported in our recent papers [12,13,14,15,16].

### 4.3. Antibacterial Activity

#### 4.3.1. MSSA and MRSA Protocols

The antibacterial activity was tested on both a methicillin-sensitive and a methicillin-resistant *S. aureus* strain, dissolving all the compounds in dimethyl sulfoxide (DMSO) and using fresh cell cultures at 10^5^ cells/mL, in a final volume of 2 mL. The detailed protocol is identical to what was previously reported [12,13,14,15,16]. Every assay was performed in quadruplicate, and for each series of experiments, both positive (no compounds) and negative (no bacteria) controls were included.

#### 4.3.2. MDRSA

The antibacterial activities on MDRSA 11.7 and MDRSA 12.1 were performed following the European Committee on Antimicrobial Susceptibility Testing (EUCAST) recommendations and Clinical and Laboratory Standards Institute (CLSI) guidelines, as detailed before [14,16]. Tetracycline was used as a positive control on every assay plate.

#### 4.3.3. Antibacterial Activity against *B. subtilis*

The two *B. subtilis* strains used were WM3612 (JH642 *amyE*::P_xyl_-*gfp-gp56*), described in Bhambani et al. [30], and WM5126 (JH642 *amyE*::P_xyl_-*gfp-zapA*), which is the same as strain FG347 [29]. For spot viability assays on plates, compounds were diluted from DMSO stocks into molten LB agar as described above and mixed well before solidifying. WM3612 cultures were grown to early exponential phase in LB broth at 37 °C, then spotted directly or after 1:10 dilution into LB onto LB plates containing various concentrations of compounds. After allowing spots to air dry, plates were then incubated overnight at 34 °C and photographed.

For fluorescence microscopy, overnight cultures of WM5126 were grown at 33 °C (after 1:200 dilution of overnight and addition of 0.1% xylose to induce expression of GFP-ZapA) for ~2 h until reaching an OD_600_ of ~0.2, then treated with no drug or with compounds 7 or 8 diluted in water from stock solutions of 40 mg/mL (8) or 27 mg/mL (7) in DMSO. Those dilutions (40 µg/mL in water) were then added at 1:100 or 1:50 to the cultures and grown for 1 h shaking at 33 °C prior to spotting on a thin layer of 1% agarose in phosphate buffered saline. Cells were imaged with differential interference contrast (DIC) and fluorescence using an Olympus BX100 fitted with a 100 × Plan Apochromat objective (N.A. 1.4) and a GFP filter set (Chroma). Images were acquired with CellSens software (Olympus) and DIC, and fluorescent images of the same field were overlaid using ImageJ [31].

### 4.4. Thiazolyl Blue Tetrazolium Bromide (MTT) Cytotoxicity Assay

Compounds showing an antibacterial activity at a concentration lower than 10 μg/mL were serially diluted in DMEM and tested for cytotoxicity on MRC-5 cells by the MTT assay (Sigma, St Louis, MO, USA). Protocol was the same previously reported [12,13,14,15,16].

### 4.5. Computational Studies

#### 4.5.1. Ligand Preparation

Compounds 1–8 required preparation, prior to further computational studies. To do so, LigPrep tool [32] included in the Schrödinger software package was used for structural preparation, as well as for 2D-to-3D conversion. All along the process, progressive levels were generated, including the addition of hydrogens, the calculation of the molecule ionization state at peculiar pH, or the generation of potential tautomers. Moreover, we evaluate, for each compound, low-energy ring conformations, and consecutive final energy minimizations, using the OPLS-2005 force field [33,34]. Aiming to mimic physiological and assay conditions, the molecules were prepared in physiological protonation states, desalted, and minimized as default in the last step.

#### 4.5.2. Molecular Properties and Predictions

Once the compounds were prepared as previously described, we analyzed the set using the Schrödinger software package, and specifically Qikprop module [35], which let us calculate, and thus, predict 44 pharmaceutical relevant ADME Tox properties, as well as undesirable properties for drug discovery. These properties include both simple molecular properties, violations in well-known drug discovery rules as Lipinski’s rule of five [36], descriptors, and relevant computational predictions for drug discovery. The results were considered to assess whether the compounds described here are in the same range of molecular properties as most known drugs and key parameters, such as lipophilicity or cellular permeability.

#### 4.5.3. Protein Preparation

As a receptor, we used the crystal structure available in the Protein Data Bank 5XDT [37,38]. The inhibitor in this crystal structure with a 2,6-difluorobenzamide scaffold identical to the moiety present in the family of inhibitors described here, which may suggest a similar binding mode in that region for this set of molecules. The protein was prepared for further computational studies following a protocol described in our previous studies [20]. As part of this, the structure of the protein was pre-processed, adding the corresponding hydrogens, avoiding the water molecules, and assigning bond orders. In addition, H-bond network assignment was made together with the calculation of the protonation states of the residues at physiological pH followed by a final restraint minimization, using the Protein Preparation Wizard [39] tool implemented on Maestro software [32].

#### 4.5.4. Induced Fit Docking

This technique, which allows changes in the active site residues geometry, to fit the ligand [40,41], was firstly performed using a reference compound in the Glide program [42] and secondly generating different poses in the active site. Moreover, Prime [43] is able to predict the protein structure, by using the pose of the corresponding ligand and through a rearrangement of nearby side chains of the active site and a minimization of the whole FtsZ energy [44]. Finally, each drug is re-docked into its corresponding low energy protein structures, and the resulting complexes are ranked according to docking score. No constraints were set, with XP (Xtra Precision) mode being used in a standard protocol, the induced fit was applied to all residues within 5.0 Å of the ligand poses, and the rest of the parameters were set as default.

#### 4.5.5. Docking Studies

Glide [44] module, in the Schrödinger software package, was employed for docking studies. We validated the protocol in our previous studies [20], and we applied it for compounds **1–8**, for conformational search, and for the evaluation of all the molecules parameters. The XP mode [45] with no constraints was applied during the docking, following the protocol and considering the same deviations recently reported [14,16].

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
