# Peer review of "Computational Design and Development of Benzodioxane-Benzamides as Potent Inhibitors of FtsZ by Exploring the Hydrophobic Subpocket"

_antibiotics, 2021, doi:10.3390/antibiotics10040442_

Round 1

Reviewer 1 Report

The manuscript entitled “Computational design and development of benzodioxane-benzamides as potent inhibitors of FtsZ by exploring the hydrophobic subpocket.” provides the computational method to explore novel FtsZ inhibiting drugs. The researchers also addressed the physicochemical and drug-like scores for those compounds. At the same time, they indicate the detailed methods to synthesize those derivatives and evaluate the efficacy in B. subtilis. At the same time, the Z ring formation has been checked after drug treatment.

However, some supporting data are missing. Several conclusions are not convincing. Further experiments or supporting data are needed.

Major Comments

  1. Line 136, Page 5 $ Figure 2. Linker for the derivatives. For this part, the author concludes that 2C linker is the optimal length. Why not show us the disadvantage of 1C and 3C linker in figure 2? At the same time, what are the states of the H-bonds for compounds 4 and 7.
  2. Line 230 & Line 237-240 Page 8. The effect of the derivatives on FtsZ. Why no show the direct effect and result for methicillin-resistant strains? subtilis is a good model. But we are more interested in the direct therapeutic effect on S. aureus. However, the author mentioned the results for S. aureus without supporting data.
  3. Figure 5.Durg disrupts Z ring formation. Except for the computational fitting for those derivatives in the pocket, could you provide some real images for the binding of the drug to the pocket and disrupting the Z ring formation.

Minor Comments

  1. Figure 4 and experimental design. As I see, we could further dilute the drug for both compounds 5 and 8 in order to reach the concentration of losing their inhibiting effect.
  2. Line 162-163, Page 6. NMR supporting data and purification. The sentence is lacking supporting data and the method to remove the unwanted derivatives.
  3. Line 220, Page8. Toxicity for MRC-5 cells. Are any other cell types used for toxicity assay?

Reviewer 2 Report

This manuscript from Straniero et al leveraged computational approach to perform docking studies using Schrödinger software package to design and develop inhibitors of FtsZ, an essential bacterial division protein. The data presented in this manuscript is a logic continuation of the authors research interests. The manuscript is well written and well designed.

Below are some of my questions for the authors.

  1. In Fig. 1, can the authors provide the information about the MIC of compound VII?
  2. In “Table 2: Inhibitory activity of compounds I, II and V and 1-8 against MSSA, MRSA, MDRSA 12.1, MDRSA 11.7 and MRC-5.” The authors should verify and confirm the absence of MRC-5, the human fibroblast cell line in the Table.
  3. Line 226 “All the compounds behave as both bacteriostatic and bactericidal, and none of them showed concerns in terms of human cytotoxicity.” It will be helpful to how the authors test the viability assay briefly. A short summary of the MTT method presented in section 4.4 could help the reader. In addition the authors should refer to the specific cell cytotoxicity data.

Minor comment

Line 656: add space between [37] and module.

In the supplementary information. The authors could change “Sommario” with “Summary”

Reviewer 3 Report

Overall, the study is well-designed and will be good contribution to the field.

In the introduction, the authors were able to establish the importance of the disease area (Staphylococcus and antibiotic resistance) but the context of the current study was not well-described. I recommend adding a short section regarding the basis of FtsZ as a target and also the benzamides as the chemical entities of interest, while citing other reviews/literature on the topic (in addition to theirs, ref 11).

As a minor comment, the manuscript will benefit with some editing to address grammatical issues (e.g., misused/misplaced articles and prepositions, incorrect past tense and plural forms of words, etc).

Round 2

Reviewer 1 Report

The authors greatly improved the manuscript.